# Human milk inhibits some enveloped virus infections, including SARS-CoV-2, in an intestinal model

Ikrame Aknouch[1,2,3,*], Adithya Sridhar[1,2,*] ⬤, Eline Freeze[1,2], Francesca Paola Giugliano[4], Britt J van Keulen[5], Michelle Romijn[5], Carlemi Calitz[1,2], Inés García-Rodríguez[1,2] ⬤, Lance Mulder[1,2], Manon E Wildenberg[4], Vanesa Muncan[4], Marit J van Gils[6], Johannes B van Goudoever[5], Koert J Stittelaar[7], Katja C Wolthers[1,†], Dasja Pajkrt[2,†] ⬤

**Human milk is important for antimicrobial defense in infants and has well demonstrated antiviral activity. We evaluated the protective ability of human milk against severe acute respiratory syndrome coronavirus 2 (SARS-CoV-2) infection in a human fetal intestinal cell culture model. We found that, in this model, human milk blocks SARS-CoV-2 replication, irrespective of the presence of SARS-CoV-2 spike-specific antibodies. Complete inhibition of both enveloped Middle East respiratory syndrome coronavirus and human respiratory syncytial virus infections was also observed, whereas no inhibition of non-enveloped enterovirus A71 infection was seen. Transcriptome analysis after 24 h of the intestinal monolayers treated with human milk showed large transcriptomic changes from human milk treatment, and subsequent analysis suggested that *ATP1A1* down-regulation by milk might be of importance. Inhibition of ATP1A1 blocked SARS-CoV-2 infection in our intestinal model, whereas no effect on EV-A71 infection was seen. Our data indicate that human milk has potent antiviral activity against particular (enveloped) viruses by potentially blocking the ATP1A1-mediated endocytic process.**

## Introduction

Viral transmission from mother to child through infected human milk is well established for viral infections such as HIV and CMV, which are known to cause perinatal disease (Ziegler et al, 1985; Stagno & Cloud, 1994; Prendergast et al, 2019). Similarly, in the case of severe acute respiratory syndrome coronavirus 2 (SARS-CoV-2), it was initially suggested that breastfeeding could potentially be a

mechanism for transmission during the acute phase of disease (Centeno-Tablante et al, 2020; Lackey et al, 2020; Trypsteen et al, 2020). However, although there have been several reports of horizontal transmission of SARS-CoV-2 to neonates, there is no direct evidence for postnatal transmission through infected human milk (Coronado Munoz et al, 2020; Han et al, 2020; Nathan et al, 2020). SARS-CoV-2 RNA has been detected in human milk from lactating SARS-CoV-2 PCR–positive mothers and viral RNA could be isolated from human milk for four consecutive days (Groß et al, 2020; Kilic et al, 2021). To date, replication competent virus has not been detected in human milk samples, although one study demonstrated that virus was culturable in Vero cells from human milk spiked with SARS-CoV-2 (Chambers et al, 2020).

The World Health Organization guidance on clinical management of COVID-19, the disease resulting from SARS-CoV-2 infection, recommends breastfeeding even for lactating mothers with a confirmed SARS-CoV-2 infection (Centeno-Tablante et al, 2020, World Health Organization, 2021). This policy seems safe as breastfeeding has been shown to reduce risk of COVID-19 among children and may prove to be critical as human milk has well demonstrated broad antimicrobial activity (Chirico et al, 2008; Ronchi et al, 2021; Verd et al, 2021). For instance, lactoferrin found in human milk exhibits antiviral effects against HIV-1, CMV, and herpes simplex virus 1 (Hasegawa et al, 1994; Harmsen et al, 1995; Berkhout et al, 2002; Seganti et al, 2004). Human milk also contains a large variety of cellular components, cytokines, and anti-inflammatory factors that may positively affect mucosal immune responses altering viral replication in target cells (Chirico et al, 2008). Most importantly, secretory immunoglobulin A (IgA) is a vital component of human milk that can confer specific protection against viral infections (Schlaudecker et al, 2013). Indeed, SARS-CoV-2–specific IgA antibodies have been detected in human milk samples and

[1]Department of Medical Microbiology, Amsterdam UMC, University of Amsterdam, Amsterdam Institute for Infection and Immunity, OrganoVIR Labs, Amsterdam, The Netherlands  [2]Department of Pediatric Infectious Diseases, Amsterdam UMC, University of Amsterdam, Vrije Universiteit, Emma Children's Hospital, Amsterdam, The Netherlands  [3]Viroclinics Xplore, Schaijk, The Netherlands  [4]Amsterdam UMC, University of Amsterdam, Amsterdam Gastroenterology Endocrinology and Metabolism, Tytgat Institute for Intestinal and Liver Research, Amsterdam, The Netherlands  [5]Department of Pediatrics, Amsterdam UMC, University of Amsterdam, Vrije Universiteit Emma Children's Hospital, Dutch National Human Milk Bank, Amsterdam, The Netherlands  [6]Department of Medical Microbiology, Amsterdam UMC, University of Amsterdam, Amsterdam Institute for Infection and Immunity, Amsterdam, The Netherlands  [7]Department of Epidemiology, Bioinformatics and Animals Models, Wageningen University, Wageningen Bioveterinary Research, Wageningen, The Netherlands

Correspondence: a.sridhar@amsterdamumc.nl; d.pajkrt@amsterdamumc.nl
*Ikrame Aknouch and Adithya Sridhar contributed equally to this work.
†Katja C Wolthers and Dasja Pajkrt contributed equally to this work.

have been shown to block SARS-CoV-2 infection in Vero cells (Fox et al, 2020; Pace et al, 2020 Preprint; Bauerl et al, 2021; van Keulen et al, 2021; Juncker et al, 2021a, 2021b). In this study, we assessed the virus neutralization potential of human milk, which either contained or lacked SARS-CoV-2 spike-specific antibodies, in a physiologically relevant primary human fetal intestinal epithelial model (Roodsant et al, 2020). We used the fetal intestinal model as a surrogate for the neonatal intestine as this is relevant for mother-to-child oral transmission and these primary cell culture models can recapitulate pathology for translational studies (Ramani et al, 2018; Sridhar et al, 2020).

# Results

### SARS-CoV-2 productively infects fetal-derived intestinal epithelial monolayers

To allow for host–pathogen interaction studies, fetal-derived intestinal organoids were expanded as 3D structures (enteroids) and opened into a 2D monolayer model using Transwell inserts as previously reported (Roodsant et al, 2020). This monolayer has a polarized layer of differentiated intestinal cells containing enterocytes, goblet, Paneth, enteroendocrine, and stem cells that recapitulates the barrier function and gene expression of the fetal intestinal epithelium. We infected these human fetal intestinal epithelial monolayers with SARS-CoV-2 and observed SARS-CoV-2 replication (Fig 1A). A 1,000-fold increase in viral RNA copies was measured by RT-quantitative (q)PCR after 96 hours post infection (hpi) in monolayers derived from two different donors. No infection was observed in the 3D enteroids possibly because of the undifferentiated state of these enteroids compared with the 2D monolayer model (Fig 1B). As observed with immunostaining of inserts that were fixed 24 hpi, infection was observed in enterocytes (Fig 1C and Video 1). Infection of enterocytes was consistent with previous reports on in vitro infection of the human intestinal epithelium with SARS-CoV-2 (Lamers et al, 2020; Zhou et al, 2020; Kruger et al, 2021).

### Human milk blocks SARS-CoV-2 infection in fetal intestinal monolayers independent of the presence of SARS-CoV-2 spike-specific antibodies, but not in Vero E6 cells

To determine the neutralizing potential of SARS-CoV-2 spike-specific antibodies in human milk, a subset of human milk samples collected as a part of the COVID MILK study was used (van Keulen et al, 2020 Preprint). This subset included human milk from two SARS-CoV-2 infected lactating mothers (diagnosed with a positive PCR from a nasopharyngeal swab and who have had clinical symptoms) and one SARS-CoV-2 PCR negative mother (at time of delivery without any COVID-19–like symptoms). SARS-CoV-2 spike-specific IgA and IgG were measured in human milk (1:10 dilution) using an ELISA. Only one of the two samples from the SARS-CoV-2 PCR–positive mothers had SARS-CoV-2 spike-specific IgG and IgA antibodies in human milk, referred to as Milk+/+ to indicate both positive nasopharyngeal swab PCR and positive human milk/

serum antibody status (Fig 2A). SARS-CoV-2 IgA and IgG spike antibodies in the human milk samples from the other SARS-CoV-2 PCR–positive mother (Milk+/−) as well as the SARS-CoV-2 PCR–negative mother (Milk−/−) were below the limits of detection. SARS-CoV-2 IgA and IgG measured in human milk samples were consistent with the SARS-CoV-2 IgA and IgG levels as measured in paired serum samples (Fig S1). In addition to the ELISA results, the neutralization potential of these human milk samples against SARS-CoV-2 was tested on Vero E6 cells. Human milk (1:20 dilution) from the three mothers was spiked with SARS-CoV-2 and incubated at 37°C for 1 h before addition to Vero E6 cells. Of the three human milk samples, only the Milk+/+ (i.e., sample containing spike-specific IgA and IgG in human milk and serum) was able to block SARS-CoV-2 infection in Vero E6 cells (Fig 2B). A detailed analysis of the ELISA results of all human milk samples in the COVID MILK study is available separately (van Keulen et al, 2020 Preprint).

To test the effect of SARS-CoV-2 IgG and IgA-containing human milk on SARS-CoV-2 replication in the human fetal intestine, the neutralization experiment was repeated on fetal intestinal epithelial monolayers derived from three different donors. On all three donor monolayers, SARS-CoV-2 infection was completely blocked by all three human milk samples (Fig 2C), demonstrating an antibody-independent antiviral activity of human milk against SARS-CoV-2. Furthermore, replication competent virus, measured at 8 h and 72 hpi, could only be isolated from the positive control (SARS-CoV-2 only) samples (Fig 2D). Human milk dilutions starting at 1:10 were used in the fetal cultures but human milk at this dilution was toxic for Vero E6 cells. Human milk dilutions of 1:20, which was the first dilution step testable on Vero E6 cells, still showed neutralization activity for all human milk samples in the intestinal cultures regardless of presence of SARS-CoV-2 IgG and IgA antibodies, albeit reduced compared with the 1:10 dilution, indicating a dose-dependent effect (Fig S2). Furthermore, to verify that the human milk was not toxic to the fetal cultures at 1:10 dilution, annexin V/propidium iodide double staining was performed to stain for necrotic or apoptotic cells. FACS analysis of the double stained intestinal cultures showed fewer dead cells in human milk-treated samples as compared with untreated samples (Fig S3), demonstrating that the human milk was not toxic to the fetal intestinal cultures at this dilution.

### Human milk blocks other enveloped RNA viruses—MERS-CoV and human respiratory syncytial virus (hRSV)-A, but does not block non-enveloped EV-A71 infection

To assess whether the human milk antiviral activity observed was SARS-CoV-2 specific or if human milk exerts a broader antiviral effect, we repeated the neutralization experiment for several other viruses. For this set of experiments, two of the human milk samples (Milk+/+ and Milk−/−) were used. Similar to SARS-CoV-2, Middle East respiratory syndrome CoV (MERS-CoV) was able to productively infect the intestinal model. Human milk (Milk+/+ and Milk−/−) was able to block the infection of MERS-CoV in the fetal intestinal cultures (Fig 3A). Next, the neutralization experiment was performed for hRSV. hRSV is an enveloped RNA virus from a different family than SARS-CoV-2 and MERS-CoV (Collins et al, 2013). Although hRSV is primarily a respiratory virus, gastrointestinal symptoms

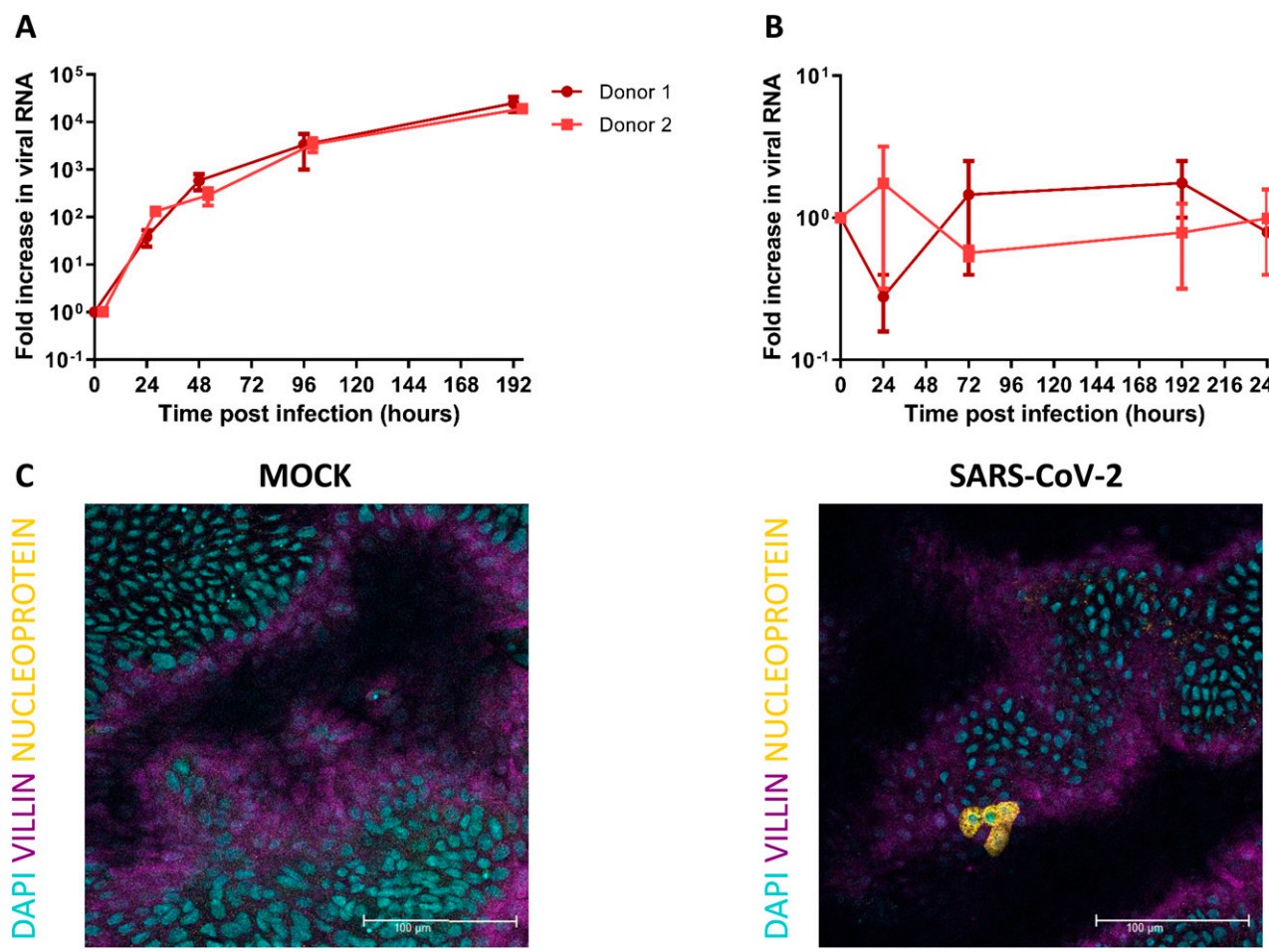

**Figure 1. SARS-CoV-2 replicates in human fetal intestinal epithelium and infects enterocytes.**
**(A, B)** Growth curves for SARS-CoV-2 replication in 2D fetal intestinal monolayer model and 3D enteroid model, respectively. Viral RNA was measured by RT-qPCR analysis of the E gene in the culture medium collected at different time points. **(C)** Immunofluorescent staining of SARS-CoV-2 infected monolayer 24 hours postinfection. SARS-CoV-2 (yellow) infects enterocytes (purple, stained for villin). Scale bars, 100 μm. Data information: In (A, B), data are presented as geometric mean ± SEM.

have also been reported (Falsey & Walsh, 2000). We observed that RSV-A can successfully infect the intestinal monolayer model. Moreover, similar to SARS-CoV-2 and MERS-CoV, human milk from the two donors (Milk+/+ and Milk−/−) was able to block RSV-A infection in the fetal model (Fig 3B). In contrast, we observed that human milk was not able to block the infection of non-enveloped enterovirus A71 (EV-A71) (Fig 3C) in this fetal model.

## Human milk elicits large changes to the fetal intestinal transcriptome

To study the effect of human milk on the fetal intestinal transcriptome, RNA sequencing (RNA-seq) was performed. Four different fetal intestinal donors were each treated for 1 h with milk only (Milk+/+, Milk+/−, and Milk−/− separately), SARS-CoV-2 pre-incubated for 1 h with milk (Milk+/+, Milk+/−, and Milk−/− separately), SARS-CoV-2 only, or were left untreated for the mock condition. The cells were lysed at 0 and 24 h post treatment (p.t.),

covering more than 1 replication cycle, and the inhibitory effect of milk shown in Fig 1C was confirmed for these samples by RT-qPCR before sequencing. As mycoplasma contamination was detected, after sequencing was complete, in other intestinal organoids during regular testing in our laboratory, the RNA-seq raw reads were also mapped to 30 reference genomes of the mycoplasma genus to assess the extent of contamination and the contaminating species in the samples used for sequencing (Langdon, 2014). *Mycoplasma hyorhinis*, a common laboratory contaminant, was detected in all samples (Fig S4). However, as all samples were contaminated and the inhibition of SARS-CoV-2 infection was confirmed on earlier (and later) organoid cultures that were negative for mycoplasma, the effect of the contamination for our question was assumed to be minimal.

Clustering analysis (Fig 4A) and principal component analysis (PCA, Fig 4B and C) showed that milk treatment of the fetal intestinal model introduced a large variation in the transcriptome over 24 h. The PCA plots clustered based on fetal donors and over time accounting for more than 60% of the variation. Differentially

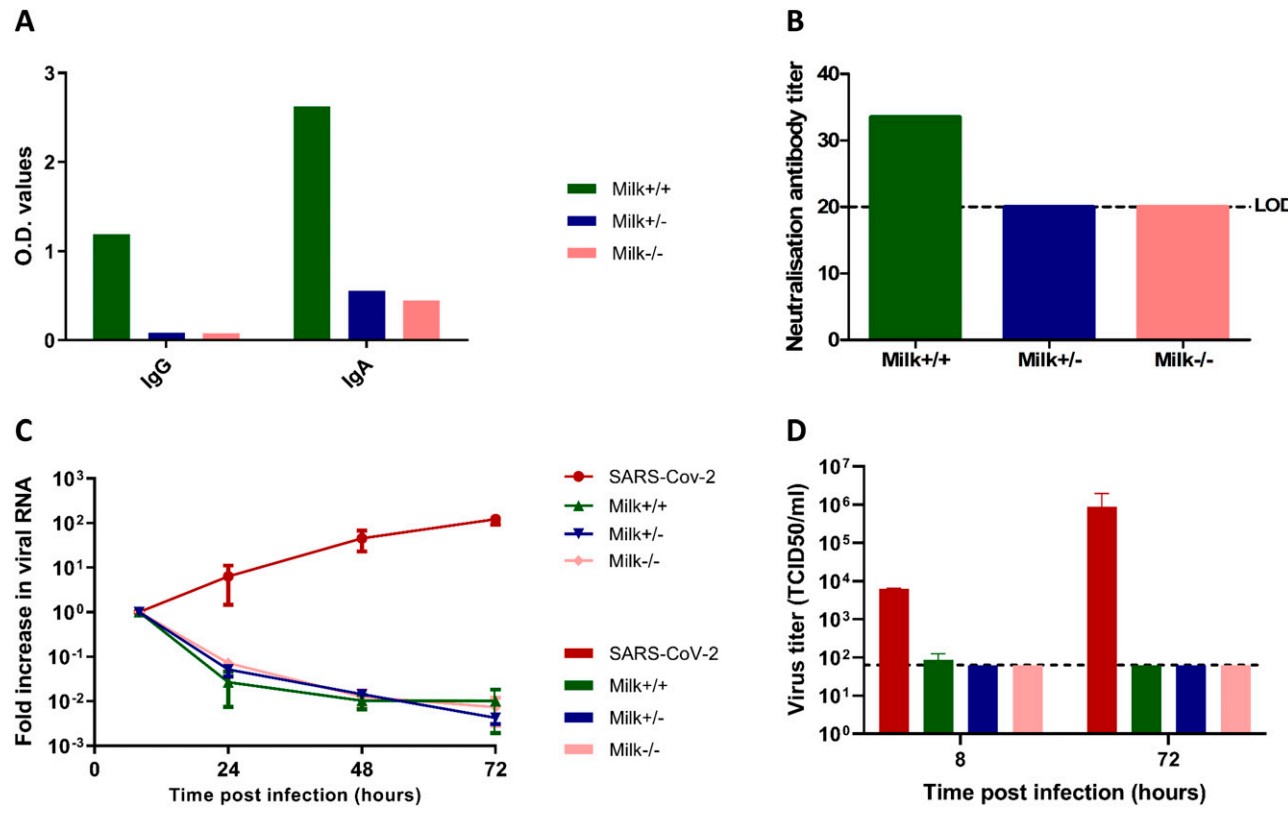

**Figure 2. Antibody titers in human milk samples and their neutralization potential in Vero E6 cells and fetal intestinal monolayers.**
**(A)** IgG and IgA titers against SARS-CoV-2 spike protein in human milk samples (1:10 dilution) collected from three lactating mothers participating in the COVID MILK study (van Keulen et al, 2020 *Preprint*), measured by ELISA. **(B)** Neutralization antibody titer of human milk measured by a neutralization assay on Vero E6 cells. Dotted lines represent the lower limit of detection. **(C)** Growth curves for SARS-CoV-2 replication under the different conditions. Each time point is a geometric mean of the values measured on three fetal-derived intestinal epithelial monolayers from three different fetal donors. SARS-CoV-2 RNA was measured by RT-qPCR analysis of the *E* gene in the culture medium collected at different time points. **(D)** Replication competent viral titers measured using a median tissue culture infectious dose (TCID50) assay. Legend: Milk+/+ was the human milk sample from a SARS-CoV-2 PCR–positive mother with SARS-CoV-2 spike-specific IgA and IgG in serum and human milk. Milk+/− was the human milk sample from a SARS-CoV-2 PCR–positive mother without SARS-CoV-2 spike-specific antibodies in serum and human milk. Milk−/− was the human milk sample from a mother negative for SARS-CoV-2 and negative for SARS-CoV-2 spike-specific antibodies in serum and human milk. Data information: In (A, B), data are presented as mean. In (C, D), data are presented as geometric mean ± SEM.

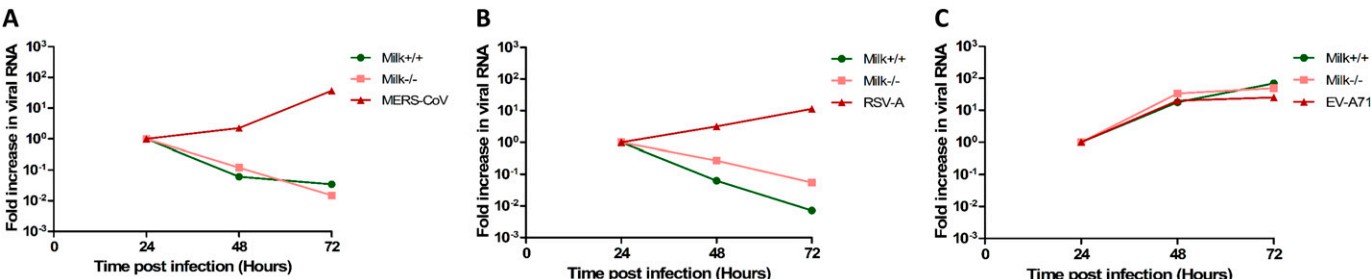

**Figure 3. Growth curves for MERS-CoV, hRSV-A, and EV-A71 replication in the fetal intestinal model under different conditions.**
**(A, B, C)** RT-qPCR analysis of viral RNA measured in medium collected at different time points. Each time point is a geometric mean of the values measured on three fetal-derived intestinal epithelial monolayers from three different fetal donors. For all three viruses, a high input was measured at the 0-h time point likely due to the viral binding to the plastic and therefore, the graphs are normalized to the 24 h time point. Data information: In (A, B, C), data are presented as geometric mean ± SEM.

expressed genes (DEGs; adjusted *P*-value < 0.01) between 0 and 24 h were depicted as volcano plots (Fig 4D–G) and highlighted large changes in the milk-treated conditions with 7,995 genes and 8,029 genes differentially expressed in milk only and milk plus SARS-CoV-2 conditions, respectively. In comparison, 2,098 genes and 124 genes were differentially expressed in the SARS-CoV-2 only and mock conditions, respectively. Gene set enrichment analysis (GSEA) showed enrichment of a wide range of pathways modulated after

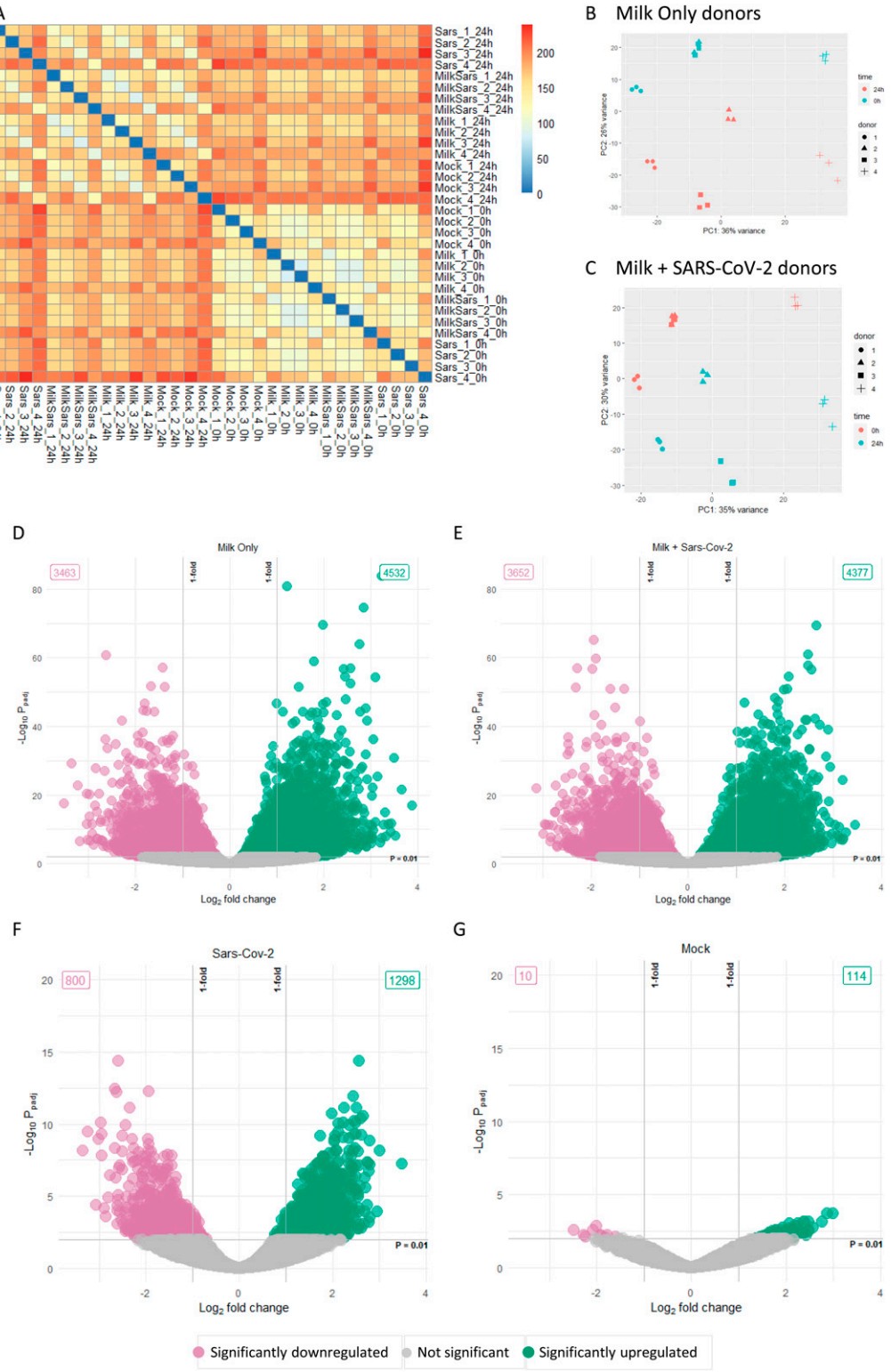

Figure 4. Overview of the transcriptome data.
**(A)** Heat map of sample-to-sample distances based on transformed count data. **(B, C)** Principal component analysis of the two milk-treated conditions. **(D, E, F, G)** Volcano plots of the log$_2$fold change against the −log$_{10}$ adjusted $P$-values of the different conditions. Significantly (adjusted $P$-value < 0.01) up-regulated and down-regulated hit shown in green and pink, respectively. Data information: In (D, E, F, G), each data point represents a gene. The log$_2$fold change of each gene is represented in the x-axis and the log$_{10}$ of its adjusted $P$-value is on the y-axis. Adjusted $P$-values are calculated using Benjamini–Hochberg procedure.

milk treatment with the top 30 shown as a dot plot (Fig S5). Furthermore, the SARS-CoV-2 treated conditions showed a statistically significant overlap of up-regulated and down-regulated genes to a dataset of SARS-CoV-2– and SARS–CoV infected enteroids (Table

S1). As the aim of the RNA-seq was to evaluate the effect of human milk on the host transcriptome, only the sole human milk-treated and human milk plus SARS-CoV-2 conditions were analyzed further.

## ATP1A1 stands out as a potential host factor in regulating SARS-CoV-2 infection

Because the DEGs list from the different human milk-treated conditions was too large to identify individual genes that might play a potential role in inhibiting SARS-CoV-2 infection with human milk, we narrowed down the list by considering the biological variation of the fetal donors. By comparing overlapping DEGs across the individual fetal donors for sole human milk and milk plus SARS-CoV-2 conditions as shown in Fig 5A–C, we identified 77 DEGs common in all comparisons (Table S2). These genes were manually screened for function and any known association with SARS-CoV-2 infection. There were several genes related to inflammation and several genes with a known association to SARS-CoV-2 infection that were differentially expressed in this list of 77 DEGs (Table 1 and Fig 5D and E). Some of the genes (*GAS6*, *IFI27*, and *SLC6A20*) were correlated to SARS-CoV-2 infection in vivo (Kasela et al, 2021; Morales et al, 2021). Three down-regulated genes (*SREBP1*, *ATP1A1*, and *KMT2C*) were reported to block SARS-CoV-2 infection when knocked out or inhibited in cell lines (Cho et al, 2020; Zhang et al, 2021). ATP1A1OS, a long non-coding RNA corresponding to the antisense RNA of ATP1A1, was also found in this list of 77 genes along with four long non-coding RNAs (RP11-113K21,4, RP11-96D1,10, RP11-498C9,13, and RP11-694I15,7). *IGF1R*, which is an entry receptor for hRSV, was also down-regulated (Griffiths et al, 2020). These 77 genes from all samples (*n* = 12) were further analyzed by evaluating their statistical significance in the least correlated (*n* = 2) and most correlated (*n* = 2) samples (Fig 5F–H and Table S2). Based on this group size effect, only 26 of the 77 genes were statistically significantly up- or down-regulated in human milk only and milk plus SARS-CoV-2 condition. Most importantly, of the three genes identified by in vitro studies, only *ATP1A1* and its antisense RNA remained significantly down-regulated (adjusted *P*-value < 0.05 in most and least correlated samples). *KMT2C* (adjusted *P*-value = 0.595) and *SREBF1* (adjusted *P*-value = 0.1235) were not significantly different in the most correlated dataset.

### Inhibition of ATP1A1 blocks SARS-CoV-2 infection in the fetal intestine

*ATP1A1* was the only gene of the 26 genes that had proven in vitro functional significance for SARS-CoV-2 infection and for the infection of several other enveloped RNA viruses including MERS-CoV and hRSV (Burkard et al, 2015; Lingemann et al, 2019; Schmidt et al, 2021). Therefore, to assess if ATP1A1 inhibition resulted in blocking SARS-CoV-2 infection in our model, the inserts were pretreated for 1 h or continuously treated (72 h) with 50 nM ouabain, a ligand of ATP1A1 (Schoner & Scheiner-Bobis, 2007). In both ouabain conditions, no SARS-CoV-2 replication (Fig 6A) was seen by RT-qPCR similar to the milk-treated conditions (Fig 2C), whereas ouabain treatment did not block the infection of EV-A71 (Fig 6B). However, light microscopy observation of the inserts showed that continuous treatment of ouabain was toxic to the cells and resulted in cell death (Fig S6). Pretreatment of ouabain for 1 h was not toxic as no cell death was seen and the cell viability in this condition was

similar to the mock inserts when compared by Annexin/PI double staining using FACS (Fig 6C).

# Discussion

Overall, our data suggest that human milk has potent antiviral effects against SARS-CoV-2 and two other enveloped viruses, namely MERS-CoV and hRSV. This antiviral activity was not dependent on the presence of SARS-CoV-2 spike-specific IgG or IgA antibodies and a lack of inhibition of non-enveloped EV-A71 replication suggests an enveloped virus specific activity. Although it is possible that there were antibodies present in the human milk samples against other SARS-CoV-2 proteins arising from natural infection, the inhibitory effects against MERS-CoV infection strongly suggest that the antiviral activity is antibody independent. MERS-CoV is an enveloped virus from the same betacoronavirus genus as SARS-CoV-2. As of January 2020 (the period when milk samples were obtained), World Health Organization reported a total of 2,519 cases of MERS-CoV infection globally and only two known cases in the Netherlands which makes it highly unlikely that human milk samples from the Dutch mothers contained MERS-CoV specific antibodies (da Costa et al, 2020; Kraaij-Dirkzwager et al, 2014). Moreover, the inhibition of SARS-CoV-2 replication in Vero E6 cells by only the spike antibody positive human milk sample further supports antibody-independent antiviral effect of human milk against these enveloped viruses in the fetal intestinal model. The lack of a broader activity in Vero E6 cells also underlines the importance of using physiological cell culture models for studying host–pathogen interactions. Low dilution (1:10) of milk was toxic to Vero E6 cells but not for the fetal intestinal model. Similar cytotoxic effects of human milk to cell lines have been previously reported and dilutions of 1:200 were required to study effects of unpurified milk on hRSV infection in HEp-2 cells (Sah et al, 2020).

The difference in neutralization activity in the different cultures also rules out a pH-sensitive inhibition of the viral envelope observed in enveloped viruses (Katow & Sugiura, 1988; Ruigrok et al, 1992). Moreover, the mean pH of human milk is estimated around 7.1 and the viral envelope is reported to remain stable at this pH (Ansell et al, 1977). This led us to hypothesize that the antiviral activity of human milk might be related to the up-regulation or down-regulation of specific host factors by human milk in the fetal cultures. It is evident from the transcriptome analysis that human milk treatment elicits large changes to the fetal intestinal transcriptome. It is important to note that the transcriptome and the proteome have varying levels of correlation and transcriptomic analysis alone does not represent a complete picture (Gunawardana & Niranjan, 2013). Furthermore, mycoplasma contamination was evidently present in all samples. However, GSEA matched with other SARS-CoV-2 datasets on intestinal organoids without this contamination and inhibitory effect of milk was verified on mycoplasma negative cultures, we assumed the influence of the contamination was minimal. It is, nonetheless, a confounding variable that is likely to affect baseline gene expression values and should be considered when interpreting our transcriptome findings (Doyle et al, 2021).

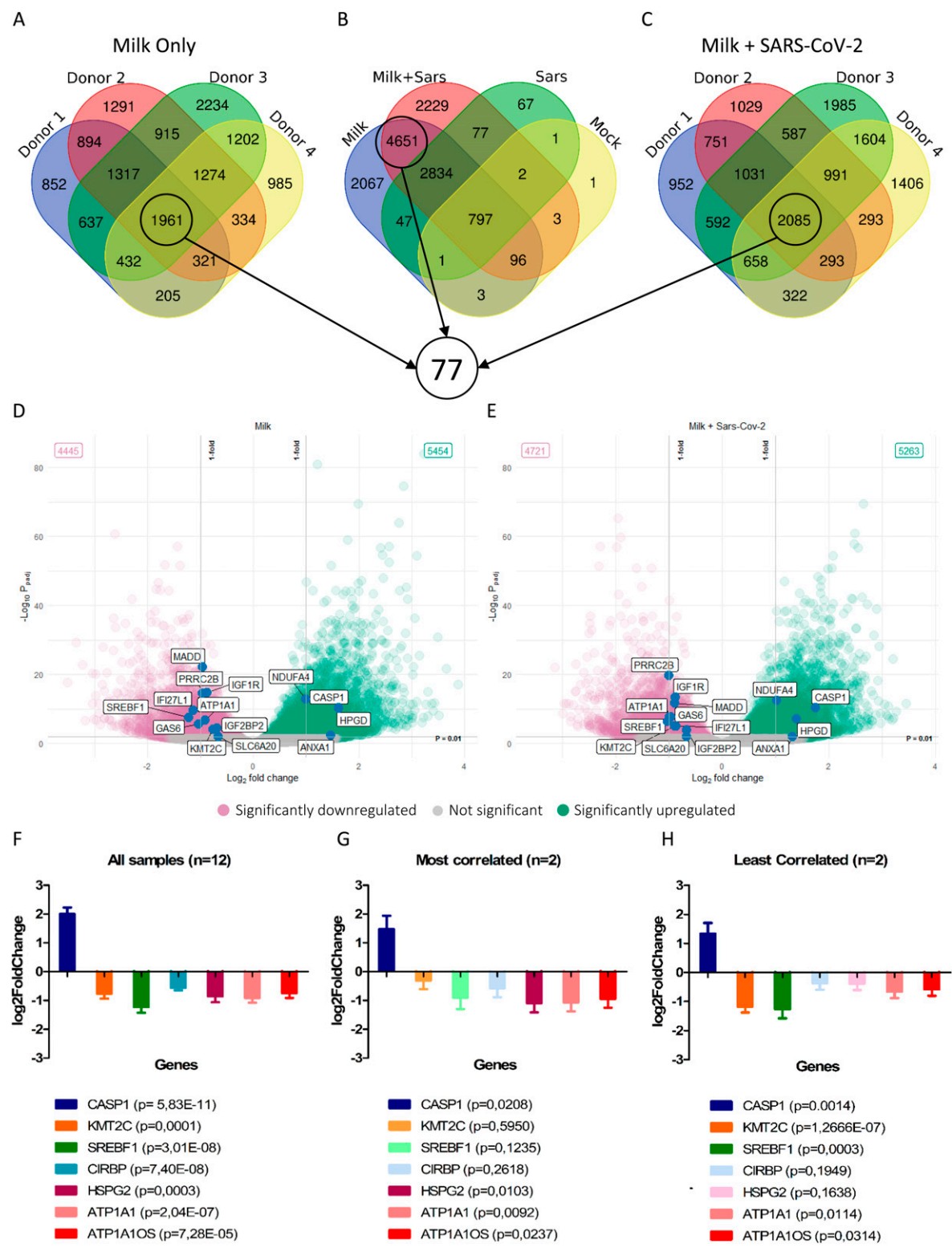

**Figure 5. Transcriptome analysis of the milk-treated conditions.**
**(A, B, C)** Venn diagram of overlapping genes across different conditions. Encircled are gene sets used for further analysis to identify common differentially expressed genes. **(D, E)** Volcano plots showing selected significant hits from the 77 genes listed in Table S2. **(F, G, H)** Log$_2$fold change values of selected differentially expressed genes from Table S2 shown for datasets containing all samples (F, n = 12), most correlated samples (G, n = 2), and least correlated samples (H, n = 2). Genes that were not significantly ($P$ > 0.05) expressed in the most correlated and least correlated datasets are faded out. Data information: In (D, E), each data point represents a gene. The log$_2$fold change of each gene is represented in the x-axis and the log$_{10}$ of its adjusted $P$-value is on the y-axis. Adjusted $P$-values are calculated using Benjamini–Hochberg procedure. In (F, G, H), data are presented as mean ± SEM. $P$-values calculated using Wald test.

**Table 1. Selected differentially expressed genes common in all milk-treated conditions.**

| Gene name | Significance |
|---|---|
| Up-regulated | |
| CASP1 | Inflammatory response initiator that results in release of mature cytokines (Feng et al, 2004). High concentrations of casp1p20 found in sera of patients (Rodrigues et al, 2021) |
| HPGD | Regulates prostaglandin E2 levels (Cho et al, 2006). SARS-CoV-2 infection reduces level of HPGD (Ricke-Hoch et al, 2021) |
| ANXA1 | Anti-inflammatory activity (Leoni et al, 2015). Serum Anxa1 levels are significantly lower in severe COVID-19 patients (Canacik et al, 2021) |
| Down-regulated | |
| SLC6A20 | Modulates COVID-19 risk (Kasela et al, 2021) |
| KMT2C | Loss of KMT2C significantly reduced infection of SARS-CoV-2 and CPE in cell lines (Baggen et al, 2021; Chan et al, 2021 Preprint) |
| IGF1R | Entry receptor for hRSV (Griffiths et al, 2020) |
| ATP1A1 | ATP1A1-mediated src signaling interferes with endocytic entry of coronaviruses and hRSV (Burkard et al, 2015; Lingemann et al, 2019) |
| IFI27L1 | Elevated IFI27 expression is associated with a high SARS-CoV-2 viral load (Shojaei et al, 2021 Preprint) |
| GAS6 | GAS6 levels reflect COVID-19 severity (Morales et al, 2021) |
| SREBF1 | SREBP1 knockdown and inhibition decreases SARS-CoV-2 viral replication (Zhang et al, 2021) |

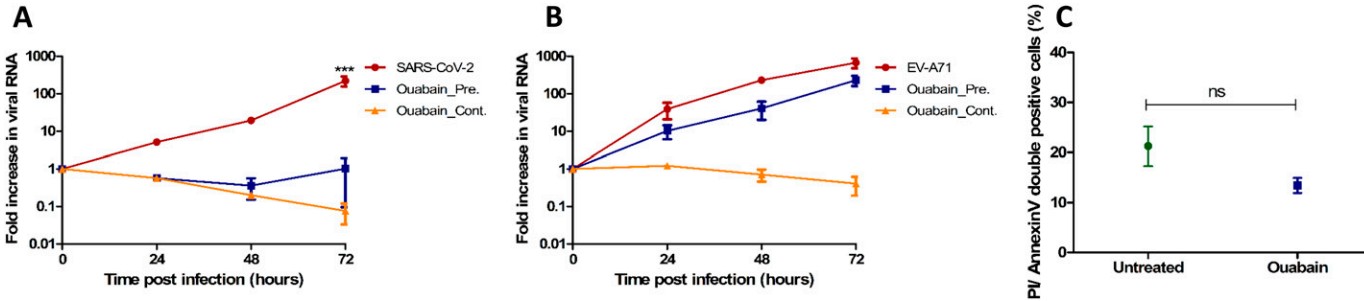

**Figure 6. Ouabain treatment of intestinal cultures and viral infection.**
**(A, B)** RT-qPCR analysis of viral RNA measured in medium collected at different time points for (A) SARS-CoV-2 infection with and without ouabain and (B) EV-A71 infection with and without ouabain. Each time point for A and B is a geometric mean of the values measured on three fetal intestinal epithelial monolayers from three different donors. **(C)** Percentage of annexin V/propidium iodide double positive cells in untreated and ouabain treated fetal intestinal cultures (n = 3 fetal donors). Differences between untreated and ouabain treated conditions were not significant. Data information: In (A, B, C), data are presented as geometric mean ± SEM. *$P \leq 0.05$ (ratio-paired $t$ test).

Whereas the GSEA analysis of pathways and gene ontology allows for a global overview of transcriptomic changes, identification of specific host factors from large DEGs list such as milk-treated samples required further refinement (Koch et al, 2018). The exploitation of the inherent biological variability between our fetal donors enabled us to narrow down DEGs list from thousands of genes to a manageable list of 77 DEGs. Although several interesting inflammatory genes and genes with known clinical or in vitro association to SARS-CoV-2 infection were identified, we were able to refine this list further through group size effects. ATP1A1, sodium/potassium-transporting ATPase subunit α-1, was the most promising candidate on this list as it had proven functional significance on the infection of all three enveloped viruses tested in our study. We were able to show that treatment with ouabain, a highly specific ligand of ATP1A1, was able to block SARS-CoV-2 infection and demonstrate this mechanism as relevant in our intestinal model. This was consistent with previous studies that showed that ATP1A1 down-regulation and ouabain treatment blocks src-mediated endocytic uptake of MERS-CoV in cell lines and hRSV in both cell lines and human airway epithelial cultures (Burkard et al, 2015; Lingemann et al, 2019). Furthermore, ouabain exhibited antiviral activity of against SARS-CoV-2 in cell lines (Cho et al, 2020). Interestingly, there are endogenous cardiotonic steroids in plasma and somatostatin, an inhibitory peptide that is known to bind to ATP1A1, is present in human milk (Werner et al, 1985; Bauer et al, 2005). However, our results only correlate ATP1A1 inhibition to antibody-independent antiviral activity of milk against several enveloped viruses and further studies into the potential mechanism regulated by specific components in human milk are necessary to validate these findings.

### Limitations of the study

The human intestinal model used in this study represents only the intestinal epithelium and the intestinal mucosa is a much more complex organ including a microbiome, mesenchymal cells, and resident immune cells that play a critical role in host–pathogen interactions. Moreover, the direct application of human milk to the

intestinal epithelium is not physiological as the milk interacts with several layers of the digestive system before reaching the intestine. As a result, the mechanism of the ATP1A1 down-regulation observed here maybe affected in a physiological setting. Finally, the intestine is not the primary replication site for the enveloped viruses tested in this study and it requires further validation if the active ingredient in human milk responsible for viral inhibition has a similar effect in the human airways.

# Materials and Methods

## Viruses and cell lines

### SARS-CoV-2
A clinical isolate of SARS-CoV-2 (BetaCoV/Munich/BavPat1/2020, passage 2) was propagated on Vero E6 monolayers (African green monkey [*Chlorocebus aethiops*] kidney, ATCC; CRL-1586) in DMEM (Gibco) supplemented with 10% FBS (Sigma-Aldrich), penicillin (100 U/ml; Lonza), and streptomycin (100 U/ml; Lonza) at 37°C. The virus stocks were produced by infecting Vero E6 cells at a MOI of 0.005 and incubating the cells for 48 h. The culture supernatant was cleared by centrifugation and stored at −80°C. Stock titers were determined by titration on Vero E6 cells. The tissue culture infectious dose 50 (TCID50) was calculated according to the method of Spearman & Kärber (Spearman, 1908; Kärber, 1931).

### MERS-CoV
MERS-CoV ($10^5$ TCID50) MERS/HCoV/KSA/EMC/2012 strain (accession no. NC_019843) was kindly provided by the Erasmus Medical Center. This isolate was propagated in Vero cells at 37°C for 3 d. The infectious virus titer was determined in Vero cells and calculated according to the Reed and Muench method (Reed & Muench, 1938).

### RSV-A
A clinical isolate of hRSV subgroup A strain was kindly provided by the Erasmus Medical Center, Rotterdam, The Netherlands. The virus was isolated in human respiratory epithelium carcinoma (HEp-2) cells from a nasal lavage of an infant hospitalized at Erasmus Medical Center, in 2011. The virus was propagated on HEp-2 cells in serum-free medium and the virus stocks were produced by infecting HEp-2 cells at a MOI of 0.01 for 2 h at 37°C, followed by addition of medium containing 10% FBS. After 24, h the serum content was reduced to 2% FBS by removing the medium and adding a serum-free medium. Cytopathic effect (CPE) was observed, and cell-free virus was harvested at ~50–75% CPE. Supernatant was clarified by centrifugation for 15 min at 1,200*g*, and subsequently aliquots were snap-frozen using a mixture of 96% ethanol and dry ice and stored at −80°C.

### EV-A71
The EV-A71 C1 91-480 was kindly provided by the National Institute for Public Health and the Environment. EV-A71 was propagated on RD99 (*Homo sapiens* muscle rhabdomyosarcoma) cells maintained in Eagle's minimum essential medium (Lonza) supplemented with 8% heat-inactivated FBS (HI-FBS; Sigma-Aldrich), 100 U/ml penicillin-streptomycin (BioWhittaker; Lonza), non-essential amino acids

(NEAA; ScienCell Research Laboratories), and 200 nM L-glutamine (Lonza). The 50% TCID50 of virus stocks was calculated according to the Reed and Muench method (Reed & Muench, 1938).

All work with infectious SARS-CoV-2 and MERS-CoV was performed in a Class II Biosafety Cabinet under BSL-3 conditions at Viroclinics Xplore and Viroclinics Biosciences-DDL.

## Isolation and culture of fetal enteroids

### Ethics statement
Human fetal intestinal tissue, gestational age 19–20 wk, was obtained by the HIS Mouse Facility of the Amsterdam University Medical Center, with written informed consents obtained from all donors for the use of the material for research purposes. The human fetal intestinal tissue was obtained with approval of the ethical committee of the Amsterdam UMC, together with approval of the experimental procedures by the HIS Mouse Facility. All experiments were performed according to the relevant guidelines and regulations, mentioned in the Amsterdam UMC Research Code.

### Culture of fetal intestinal organoids
For the generation of fetal small intestinal organoids, crypts were isolated from fetal intestinal tissue as described previously (Sato et al, 2009). Isolated crypts were suspended in Matrigel (Corning), dispensed in three 10 $\mu$l droplets per well in a 24-well tissue culture plate and covered with 500 $\mu$l medium. Enteroid cultures were maintained in Human IntestiCult Organoid Growth Medium (HIOGM, STEMCELL Technologies) with 100 U/ml penicillin–streptomycin (Gibco, Thermo Fisher Scientific) and incubated at 37°C, 5% $CO_2$. Medium was changed every 2–3 d and organoids were passaged every 6–10 d as described previously (Roodsant et al, 2020) or by enzymatic dissociation as described below for seeding onto cell culture inserts.

### Intestinal epithelial monolayer culture
The fetal intestinal epithelial monolayers were cultured on Transwell cell culture inserts (6.5 mm, 3.0 $\mu$m pore size; VWR) as described previously (Roodsant et al, 2020). Briefly, the inserts were coated with 100 $\mu$l of 20 $\mu$g/ml collagen type I (rat tail, Ibidi) in 0.01% (vol/vol) acetic acid (Sigma-Aldrich) for 1 h at RT, and washed with PBS (Lonza) before use. Human fetal enteroids were collected, and a single cell suspension was obtained by treatment with TrypLE (Gibco, Thermo Fisher Scientific) for 10 min at 37°C. Cells were diluted to $10^6$ cells/ml and 100 $\mu$l of cell suspension per insert was seeded ($10^5$ cells per insert). The cells were then cultured and differentiated over a period of 14 d as described previously (Roodsant et al, 2020).

### Infection of the fetal 3D enteroids with SARS-CoV-2
The fetal enteroids were harvested in ice cold advanced DMEM/F12 supplemented with 100 U/ml penicillin/streptomycin, 7.5 mM Hepes, and 0.5× GlutaMAX, washed once to remove the Matrigel, and sheared using TrypLE. After shearing, enteroids were washed once in cold Advanced DMEM/F12 before infection with $10^4$ PFU of SARS-CoV-2. After 1-h incubation of the virus at 37°C, the enteroids were washed three times with advanced DMEM/F12 to remove unbound virus. Enteroids were re-embedded into 10 $\mu$l Matrigel in

24-well tissue culture plates and cultured at 37°C. At 0, 24, 192, and 240 hpi, a 500 $\mu$l sample was taken and the collected volume was replaced with fresh culture medium.

### Viral infection of fetal intestinal monolayers
The fetal intestinal monolayers were infected apically with $10^4$ PFU of virus. After 1-h incubation at 37°C, unbound virus was removed by washing with PBS and fresh medium was added. At different time points, 100 $\mu$l sample was taken from the apical compartment and the collected volume was replaced with fresh differentiation medium.

### Neutralization assay using Vero E6 cell monolayers
The presence of SARS-CoV-2 neutralizing antibodies in the human milk and sera was tested using a virus neutralization assay. Working dilution containing ~400 TCID50/well of SARS-CoV-2 was mixed with 60 $\mu$l of serial twofold dilutions of heat-inactivated serum or human milk in triplicates and incubated for 60 min at 37°C. Subsequently, 100 $\mu$l of the virus/sera and virus/human milk mixtures were added to confluent Vero E6 monolayers and incubated for 5 d at 37°C after which plates were scored using the vitality marker WST-8 (colorimetric readout; Sigma-Aldrich). To this end, WST-8 stock solution was prepared and added to the plates. Per well, 20 $\mu$l of this solution (containing 4 $\mu$l of the ready-to-use WST-8 solution from the kit and 16 $\mu$l infection medium, 1:5 dilution) were added and incubated for 5 h at RT. Subsequently, plates were measured for optical density at 450 nm (OD450) using a micro plate reader. A titer of ≥20 was considered to be positive.

### Human milk treatment of fetal intestinal monolayer against SARS-CoV-2 infection
To evaluate the protective efficacy of human milk against SARS-CoV-2, the human milk samples were diluted 1:5 and mixed 1:1 with a working dilution containing SARS-CoV-2 ($10^5$ TCID50/100 $\mu$l) in duplicates. The virus/human milk mixture was incubated for 60 min at 37°C. Subsequently, 100 $\mu$l of the virus/human milk mixtures were added apically to the fetal intestinal monolayer. After 1 h incubation, unbound virus was removed, and fresh medium was added. At 8, 24, 48, and 72 hpi, a 100 $\mu$l sample was taken from the apical compartment and the collected volume was replaced with fresh medium.

### Immunostaining and immunofluorescence microscopy
Undifferentiated fetal enteroids were seeded on human collagen–coated $\mu$-slide eight well chambered coverslips (iBIDI). At 72 h post seeding, differentiation was induced by adding cell differentiation medium. Upon full differentiation, enteroids were infected with SARS-CoV-2. At 24 hpi, the cells were fixed with 4% formaldehyde for 30 min at RT. After fixation, the cells were washed twice with PBS, then permeabilized with 0.5% Triton X-100 for 5 min. Cells were blocked in BlockAid Blocking Solution (Thermo Fisher Scientific) for 1 h at RT. Cells were then washed twice with washing solution (Image-iT Fixation/Permeabilization Kit; Thermo Fisher Scientific) and incubated with SARS-CoV-2 Nucleocapsid (Thermo Fisher Scientific) and Villin (Santa Cruz Biotechnology) antibodies in blocking solution (Image-iT Fixation/Permeabilization Kit; Thermo Fisher Scientific) containing 3% BSA. After washing three times with

PBS, cells were incubated with Alexa-conjugated secondary antibodies goat anti-rabbit Alexa Fluor 647 and goat anti-mouse Alexa Fluor 488 (Thermo Fisher Scientific) in PBS blocking solution (Image-iT Fixation/Permeabilization Kit; Thermo Fisher Scientific) containing 3% BSA for 1 h at RT. Cells were washed three times with PBS and incubated with ReadyProbes Tissue Autofluorescence Quenching Mix (Thermo Fisher Scientific) for 5 min at RT. After washing three times with the washing solution (Image-iT Fixation/Permeabilization Kit; Thermo Fisher Scientific), the cells were incubated with DAPI (Invitrogen). The iBIDI chambers were imaged on a Leica TCSS SP8 X mounted on a Leica DMI6000 and analyzed using LAS X (Leica Microsystems), Huygens Professional software (Scientific Volume Imaging), and ImageJ (version 1.52a; National Institutes of Health).

### Viral RNA isolation and RT-qPCR
RNA extraction was performed using a MagNA Pure 96 (Roche Diagnostics Nederland B.V.) with MagNA Pure 96 DNA and Viral NA Small Volume Kit with an input volume of 25 $\mu$l and output volume of 100 $\mu$l according to the manufacturer's instructions (Roche Diagnostics Nederland B.V.). The extraction was internally controlled by the addition of a known concentration of phocine distemper virus. A volume of 8 $\mu$l extracted RNA was amplified using a 7500 Realtime PCR System (Applied Biosciences) in a 20 or 50 $\mu$l final volume, containing the TaqMan Fast Virus 1-Step Master Mix (Thermo Fisher Scientific) and primers and probe mixture for detection of the virus and phocine distemper virus in a duplex reaction. The number of virus copies in the different samples was calculated using the resulting Ct value against a standard curve. Primer sequences are listed in Table S3.

### Virus titration
The presence of SARS-CoV-2 infectious viral particles in the collected medium samples was detected by virus titration on Vero E6 monolayers. Quadruplicate 10-fold serial dilutions were performed in confluent monolayers of Vero E6 cells and incubated for 1 h at 37°C. Vero E6 monolayers were washed and incubated for 5 d at 37°C, after which plates were scored and the TCID50 was calculated according to the Spearman & Kärber method (Spearman, 1908; Kärber, 1931).

### Annexin V/propidium iodide cell apoptosis assay
To verify that human milk is not toxic to fetal cultures at 1:10 dilution, the fetal enteroid monolayers were cultured on Transwell cell culture inserts. Human milk was added apically to the monolayers and incubated for 1 h at 37°C. Human milk was then removed, and fresh medium was added. After incubation for 24 h at 37°C, a single-cell suspension was obtained by treatment with TryplE (Gibco, Thermo Fisher Scientific) for 10 min at 37°C. Subsequently, the cells were centrifuged for 5 min at 300$g$ and the cell pellets were washed with 10% FBS in PBS. The cells were then incubated in the binding buffer (BD Biosciences) with annexin V (BD Biosciences) and propidium iodide (Sigma-Aldrich) for 15 min at RT. After the incubation, annexin V/propidium iodide double stained cells were measured simultaneously using Flow Cytometry (FACS LSRFortessa; BD Biosciences) and data obtained were analyzed using the FlowJo software (Treestar).

### COVID MILK study

This prospective case control study included lactating women with a confirmed or high probability of a SARS-CoV-2 infection and healthy controls with a negative SARS-CoV-2 PCR during delivery and without any symptoms of COVID-19 (van Keulen et al, 2020 Preprint). Participants were requested to collect 100 ml of human milk and to store it in their freezer until it was collected during a home visit. Subsequently, the samples were stored at –20°C. Blood was also collected from these mothers to measure serum antibody levels. All samples were obtained with written informed consent for use of the material for research purposes.

### ELISA for antibodies in human milk and serum

The binding capacity of antibodies (IgG and IgA) in serum and milk to SARS-CoV-2 spike were measured using an ELISA assay, as previously described (Brouwer et al, 2020). In brief, soluble perfusion-stabilized S-protein of SARS-CoV-2 using stabilization strategies were generated as previously described (Wrapp et al, 2020 Preprint). This protein was immobilized on a 96-well plate (Greiner Bio-One) at 5 $\mu$g/ml in 0.1 M NaHCO$_3$ overnight, followed by a 1 h blocking step with casein (Thermo Fisher Scientific). Human milk was diluted 1:5 and serum were diluted 1:100 in casein and incubated on the S-protein–coated plates for 2 h to allow binding. Antibody binding was measured using 1:3,000 diluted HRP-labeled goat anti-human IgG (Immunoresearch) in casein for the serum samples and 1:3,000 diluted HRP-labeled goat anti-human IgA (BioLegend) in casein for the human milk samples.

### Statistical analysis

Statistical analysis was performed, and graphs were generated using Prism (version 8.3.0; GraphPad Software). Enteroids derived from three different donors were used for all experiments to obtain three biological replicates, unless state otherwise. All experiments were performed in duplicates for each donor and geometric mean of biological replicates was plotted, unless stated otherwise, as indicated in figure legends. Differences between milk-treated and untreated conditions were tested with a ratio-paired $t$ test, with $P$-value < 0.05 considered statistically significant.

## RNA sequencing and transcriptome analysis

### Fetal intestinal cultures for transcriptome analysis

Human milk samples were diluted 1:5 and mixed 1:1 with working dilution of SARS-CoV-2 (10$^5$ TCID50/100 $\mu$l). The virus/human milk mixtures, milk only, and SARS-CoV-2 samples were incubated for 1 h at 37°C. Subsequently, 100 $\mu$l of either the virus/human milk mixture, human milk, SARS-CoV-2, or media was added apically to four fetal intestinal monolayers. After 1 h incubation, the monolayers were washed three times with PBS (Lonza). At 0 and 24 hpi, the monolayers were lysed, and RNA was extracted for RT-qPCR and sequencing. Before sequencing, the viral RNA was measured in all samples as described earlier.

### RNA sequencing

RNA sequencing of the isolated viral RNA was performed by GenomeScan BV. In brief, samples were prepared using NEBNext Low Input RNA Library Prep Kit for Illumina (New England BioLabs),

as per the manufacturer's instructions. Quality and yield after sample preparation were measured with the Fragment Analyzer (Illumina) and size of the resulting products was consistent with expected size distribution (a peak between 200 and 400 bp). Clustering and DNA sequencing was performed, according to the manufacturer's instructions, using the NovaSeq600 (Illumina). Image analysis, base calling, and quality check was performed with the Illumina data analysis pipeline RTA3.4.4 and Bcl2fastq v2.2.

### Data analysis

Raw reads of the sequencing data were first run through a quality control check using the QC tools FastCQ v0.11.9 and FastQA v3.1.25. Samples containing low-quality, adapter-polluted, or high content of unknown bases are removed using Fastp v0.20.1. The cleaned up reads were mapped against 30 references of the Mycoplasma genomes, using FastQ Screen v0.13.0, and against the human GRCh37.75 (HomoSapiens.GRCh37.75.dna.primary_assembly.fa) using STAR v2.5.4 in default settings. The feature count of the unique reads within exon regions were counted by using the mapping locations. Only the unique reads within the exon regions were counted. The differential expression was determined with DESeq v1.5 and DESeq2 v2-1.14. Differential expressed genes with a $P$-adjusted of <0.05 were classified as significant. The heat maps, PCA plots, and volcano plots were generated with R v.4.1.1 packages ggplot2 v3.3.5 (Wickham, 2011). The IDs of all genes were selected from the significant gene lists and Venn diagrams were created using a web tool (https://bioinformatics.psb.ugent.be/webtools/Venn/) from the University of Gent. A correlation matrix was created from all milk only and milk plus SARS-CoV-2 samples ($n$ = 12) using R. From this matrix, milk samples with the lowest and highest correlation ($n$ = 2) were analyzed with DESeq2 v2-1.14 to generate DEGs from these datasets.

### GSEA

GSEA analysis of the milk-treated samples was performed using default settings of a web-based platform, NASQAR running clusterProfiler (Yu et al, 2012; Yousif et al, 2020; Wu et al, 2021). KEGG dot plots and text files of enriched GO-terms and KEGG pathways were saved. GSEA analysis of the up-regulated and down-regulated genes from the SARS-CoV-2 DEG list was performed using Enrichr and compared with COVID-19–related gene sets (Chen et al, 2013; Kuleshov et al, 2016, 2020; Xie et al, 2021).

### Effect of ouabain treatment on SARS-CoV-2 infection in fetal intestinal monolayers

To evaluate the role of ATP1A1 inhibition on SARS-CoV-2 infection in the fetal intestinal monolayers, ouabain was used. For the pretreatment condition, the cultures from three different donors were pretreated with 50 nM ouabain (Sigma-Aldrich) in organoid culture media 1 h before infection at 37°C. After 1 h, the medium containing ouabain was removed and fresh medium containing SARS-CoV-2 or EV-A71 (10$^5$ TCID50/100 $\mu$l) was added apically. For the continuous treatment condition, no pretreatment was performed and 50 nM ouabain was added to organoid culture medium during the whole course of infection. Samples were collected at different time points and viral RNA was measured by RT-qPCR. To test for toxicity of ouabain treatment, additional control cultures were used, and cell viability was measured by FACS as described earlier.

## Data Availability

The RNA sequencing data from this publication have been deposited to Annotare and assigned the identifier "E-MTAB-11979."

### Ethics statement

The use of fetal material is determined by Dutch law (Wet Foetal Weefsel) that human fetal tissues/cells can only be used for medical purposes, medical and scientific research, and medical and scientific education. All material has been collected from donors from whom a written informed consent has been obtained for the use of the material for research purposes. The fetal donor information is anonymized and is not available to the Amsterdam UMC. The breastmilk samples were obtained with written informed consent from all participants as a part of the COVID MILK study registered in the Dutch Trial Register (NL 8575) on 1 May 2020. Ethics approval was obtained from the Medical Ethics Committee of the Amsterdam UMC.

## Supplementary Information

## Acknowledgements

HIS mouse facility (Amsterdam UMC, The Netherlands) is acknowledged for providing fetal tissues. The authors would like to thank Dr. Kees Weijer, Mrs. Esther Siteur-van Rijnstra, Mrs. Cynthia A van der Linden, and Dr. Arie Voordouw for facilitating the provision of the fetal material. Cellular Imaging core facility of the Amsterdam UMC, The Netherlands is acknowledged for the advanced light microscopy. The authors also wish to thank Jonneke de Rijck, Guido van der Net, and Lie Mulder from Viroclinics Xplore, Gerrit Koen, Hetty van Eijk, and Thomas J Roodsant from the Department of Medical Microbiology, Amsterdam UMC, Jacqueline Vermeulen from the Tytgat Institute, Amsterdam UMC, and Daisy I Picavet-Havik and Ron A Hoebe from the Cellular Imaging core facility, Amsterdam UMC for their technical support. This work was funded under the OrganoVIR project (grant 812673) and GUTVIBRATIONS (grant 953201) in the European Union's Horizon 2020 programme, the PPP allowance made available by Health~Holland, Top Sector Life Sciences and Health, to Amsterdam UMC, location Academic Medical Center to stimulate public-private partnerships, and funding from Stichting Steun Emma Kinderziekenhuis. The funders had no role in the design of the study, data analysis, writing of the manuscript, or in the decision to publish the results.

### Author Contributions

I Aknouch: conceptualization, formal analysis, investigation, visualization, methodology, and writing—original draft.
A Sridhar: conceptualization, formal analysis, funding acquisition, investigation, visualization, methodology, and writing—original draft.
E Freeze: formal analysis, visualization, and writing—original draft.
FP Giugliano: formal analysis, investigation, and writing—review and editing.
BJ van Keulen: funding acquisition, investigation, and writing—review and editing.
M Romijn: investigation and writing—review and editing.
C Calitz: formal analysis and writing—review and editing.
I García-Rodríguez: investigation and writing—review and editing.
L Mulder: investigation and writing—review and editing.
ME Wildenberg: resources, supervision, and writing—review and editing.
V Muncan: resources, supervision, funding acquisition, and writing—review and editing.
MJ van Gils: resources, supervision, investigation, and writing—review and editing.
JB van Goudoever: resources, supervision, funding acquisition, and writing—review and editing.
KJ Stittelaar: conceptualization, resources, supervision, funding acquisition, methodology, and writing—review and editing.
KC Wolthers: conceptualization, resources, supervision, funding acquisition, methodology, writing—original draft, and project administration.
D Pajkrt: conceptualization, resources, supervision, funding acquisition, methodology, writing—original draft, and project administration.

### Conflict of Interest Statement

I Aknouch is an employee of Viroclinics Xplore. JB van Goudoever is a founder and director of the Dutch National Human Milk Bank and member of the National Health Council. JB van Goudoever has been a member of the National Breastfeeding Council from March 2010 to March 2020. The other authors declare no competing interests.

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
