## [Reviewer comments · Life Science Alliance]

Life Science Alliance

Human milk inhibits some enveloped virus infections, including SARS-CoV-2, in an intestinal model

Ikrame Aknouch, Adithya Sridhar, Eline Freeze, Francesca Giugliano, Britt van Keulen, Michelle Romijn, Carlemi Calitz, Inés García-Rodríguez, Lance Mulder, Manon Wildenberg, Vanesa Muncan, Marit van Gils, Johannes van Goudoever, Koert Stittelaar, Katja Wolthers, and Dasja Pajkrt

DOI: <https://doi.org/10.26508/lsa.202201432>

Corresponding author(s): Adithya Sridhar, Amsterdam University Medical Centers

Review Timeline:

Submission Date:	2022-03-01
Editorial Decision:	2022-05-20
Revision Received:	2022-05-31
Editorial Decision:	2022-06-27
Revision Received:	2022-07-13
Accepted:	2022-07-14

Scientific Editor: Novella Guidi

Transaction Report:

May 20, 2022

Re: Life Science Alliance manuscript #LSA-2022-01432-T

Dr. Adithya Sridhar
Amsterdam UMC
Netherlands

Dear Dr. Sridhar,

Thank you for submitting your manuscript entitled "Human milk inhibits enveloped virus infections, including SARS-CoV-2, in an intestinal model" to Life Science Alliance. The manuscript was assessed by expert reviewers, whose comments are appended to this letter. We invite you to submit a revised manuscript addressing the Reviewer comments.

Thank you for this interesting contribution to Life Science Alliance. We are looking forward to receiving your revised manuscript.

Sincerely,

B. MANUSCRIPT ORGANIZATION AND FORMATTING:

Reviewer #1 (Comments to the Authors (Required)):

This report is interesting and provides an interesting viewpoint on the impact of human milk on viral replication in the gut. Based on the data provided, however, the conclusions are overstated, and the title is somewhat misleading and should be changed.
Comments:

In Fig.1C, infection (yellow area) seems very limited/localized, please explain. Was amount of virus used for these experiments sufficient? What if replication was more pronounced, would milk still have the same effect?

p.4, line 137 - where do you show that RSV replicates in this assay?

There is a big claim here that milk doesn't inhibit non-enveloped viruses, but only 1 was tested (and only 3 enveloped viruses, 2 of which that are highly related). If this claim is to be so strongly made, then more viruses need to be tested in this model.

The possibility that antibodies against other SARS-CoV-2 antigens are present in the milk is mentioned, but why was that not tested? this would be a simple test, and should be done.

slgA has non-specific anti-viral properties, would be very informative to repeat experiments with antibodies depleted.

Authors should discuss the mechanism whereby some kind of inhibitor of ATP1A1 in the milk would block virus replication.

Fig. 2B: This is with milk, which has high background in this assay, was purified IgA tested?

Fig 2A,B: was this a single replicate? Was a lower dilution tested in the ELISA like 1/2, as has been reported for this assay previously?

Fig. 3: these titers are not at all convincing in terms of actual replication? looks more like cells are just dying vs. not dying? please discuss further as these data are not convincing compared to SARS-CoV-2...

Reviewer #3 (Comments to the Authors (Required)):

The study by Aknouch and Sridhar et al evaluated the potential protective ability of human milk against SARS-CoV-2 infection in a human fetal intestinal cell culture model. The study is well performed and provide valuable information not only to the SARS-CoV-2, but also to other viruses like RSV where lung-gut axis play an important role.

Major Comments:

1. Fig 1A/B: This is one interesting observation. It has been shown that SARS-CoV-2 can infect several cells/tissue but the infection is abortive. Can author perform the viral RNA detection in the cells of 3D enteroid model? if so it can be valuable to discuss this in the discussion section.

2. In the transcriptomics I really appreciate the honesty of the authors to report the mycoplasma contamination which most of the time unreported. However amount of the mycoplasma can be directly associated with the signaling cascade. Though authors reported DEG with or without some samples, bit detailed analysis of the Fig S4 as more supplementary files, e.g. what are the overlap of genes, did the key 77 genes changes after changing the removal of the most significant genes?

3. The top key gene ATP1A1 seems to have role in AKT pathway and it was shown earlier that SARS-CoV-2 can uses these pathways for it's replication. Please discuss about that.

4. Please add limitation section in the manuscript.

Minor Comments:

1. Line 180-181: It states "Some of the SARS-CoV-2 associated genes (GAS6, IFI27, SLC6A20)" Please change the terms, they are not SARS-CoV-2 associated genes, these are for many viruses e.g IFI27.

Reviewer 1

We thank the reviewer for reviewing our manuscript and we appreciate the concerns raised.

Reviewer 1, comment 1

In Fig.1C, infection (yellow area) seems very limited/localized, please explain. Was amount of virus used for these experiments sufficient? What if replication was more pronounced, would milk still have the same effect?

Our response

A MOI of 0.2 was used in the organoid experiments and the replication of SARS-CoV-2 in the intestinal model is consistent with those reported in literature. For instance, Lammers et al., 2020 observed a 2-log and 3-log increase in infectious particles over 24h and 48h, respectively [1]. Similarly, Zhao et al., 2021 observed a 2-log and 3-log increase in copy numbers over 24h and 48h, respectively [2]. Based on the work by Zhao et al., it appears that SARS-CoV-2 replicates less efficiently than SARS-CoV-1 in the intestinal epithelium. This is likely due to a strong innate immune response in these organoid cultures as Stanifer et al., 2020 were able to generate higher titers after blocking the JAK-STAT pathway [3]. A small cluster of cells is seen in Fig. 1C as this represents only a thin section on the confocal microscope taken at an early time point (24h) and is, therefore, not indicative of poor replication. Moreover, as seen with the enterocyte staining (villin), the cultures are not flat and therefore, the virus particles appear in different planes. For clarification, we added a video of the z-stack to show the presence of virus on other planes.

Reviewer 1, comment 2

p.4, line 137 - where do you show that RSV replicates in this assay?

Our response

In the set of experiments described in Figure 3, there was an issue with high input despite several wash steps. Therefore, after normalization, the change in viral RNA is not immediately apparent. When comparing the virus-only condition against the two milk-treated conditions, viral RNA increases after 24h in the virus-only condition while the milk-treated conditions continue to drop (except for EV-A71 where milk has no inhibitory effect). Based on this, we concluded that RSV-A was replicating despite the high input levels measured at the 0h time point. To make it more clear, we have normalized this figure to the 24h time point instead of 0h.

Reviewer 1, comment 3

There is a big claim here that milk doesn't inhibit non-enveloped viruses, but only 1 was tested (and only 3 enveloped viruses, 2 of which that are highly related). If this claim is to be so strongly made, then more viruses need to be tested in this model.

Our response

We fully agree with the reviewer that we cannot conclude that milk inhibits enveloped viruses in general and has no effect against non-enveloped viruses. We have specifically stated in the title that milk inhibits infection of some enveloped viruses. Moreover, it is stated in the abstract that milk has potent antiviral activity against particular (enveloped) viruses with the word enveloped in brackets

to avoid any broad conclusions. On the portal, we had to shorten our title due to character limits and the word “some” was left out but this has been corrected in the resubmission.

Reviewer 1, comment 4

The possibility that antibodies against other SARS-CoV-2 antigens are present in the milk is mentioned, but why was that not tested? this would be a simple test, and should be done.

Our response

The human milk samples used in our study are a part of a larger cohort (COVID MILK Study) looking at SARS-CoV-2 antibodies in human milk [5]. In the COVID MILK study, antibodies against spike, receptor binding domain, and nucleocapsid (N) protein were tested in the serum. The antibody negative milk samples used in our study were negative for both the spike and N protein. This has been added to the supplementary figure S1 captions. Moreover, it is reported that the majority of neutralizing antibodies against SARS-CoV-2 targets the receptor binding domain of the spike protein [6]. Furthermore, the lack of any neutralization activity in the Vero cells indicates that the antiviral effect of human milk is antibody independent.

Reviewer 1, comment 5

sIgA has non-specific anti-viral properties, would be very informative to repeat experiments with antibodies depleted.

Our response

We agree with the reviewer that the depletion of antibodies will be an informative experiment. However, in line with the previous comment, our data strongly suggests that the antiviral effect of human milk is through the regulation of host factors rather than direct action on the virus. Thus, while it is possible IgA has a non-specific antiviral effect on the host, this set of experiments will dive into the exact factor behind the antiviral effect and will also require studying other antiviral factors in the human milk extending the scope of our study beyond its current limits.

Reviewer 1, comment 6

Authors should discuss the mechanism whereby some kind of inhibitor of ATP1A1 in the milk would block virus replication.

Our response

We have alluded to potential inhibitors (cardiotonic steroids and somatostatin) of ATP1A1 being present in human milk in the discussion on lines 279-281. There are other possibilities such as calcium in milk leading to excess extracellular calcium and downregulation of related pathways resulting in ATP1A1 downregulation. However, we feel this hypothesis is too speculative and has, therefore, been left out.

Reviewer 1, comment 7

Fig. 2B: This is with milk, which has high background in this assay, was purified IgA tested?

Our response

The antibody titer in Figure 2B for the negative samples is not high background but indicates the limit of detection of the assay. As stated in the figure, anything below the dotted lines is below the

limit of detection. Figure 2B has been modified to indicate that the dotted lines represent the limit of detection of the assay.

Reviewer 1, comment 8

Fig 2A,B: was this a single replicate? Was a lower dilution tested in the ELISA like 1/2, as has been reported for this assay previously?

Our response

Both the ELISA and neutralization assays were performed in triplicates. For the ELISA, human milk was diluted 1:5 and the serum was diluted 1:100. This is mentioned in the methods section on lines 482-483.

Reviewer 1, comment 9

Fig. 3: these titers are not at all convincing in terms of actual replication? looks more like cells are just dying vs. not dying? please discuss further as these data are not convincing compared to SARS-CoV-2...

Our response

As mentioned earlier, there was a high background despite several rounds of washing in these assays. In the SARS-CoV-2 experiments, an 8-hour time point was included to capture the first replication cycle but this was not included in the subsequent experiments due to increased restrictions within the BSL-3 facility. These graphs have now been normalized to the 24h time point and the caption has been amended to explain this normalization. The effect of the human milk on the replication of these viruses is evident as the viral RNA increases in the virus-only condition while it drops steadily in the virus plus milk conditions.

Reviewer 3

We thank the reviewer for the feedback and for their kind words in highlighting the importance and novelty. The specific comments are addressed below.

Reviewer 3, comment 1

Fig 1A/B: This is one interesting observation. It has been shown that SARS-CoV-2 can infect several cells/tissue but the infection is abortive. Can author perform the viral RNA detection in the cells of 3D enteroid model? if so it can be valuable to discuss this in the discussion section.

Our response

We agree with the reviewer that the lack of replication in 3D is interesting. However, we do not believe it is because the infection is abortive in the 3D system. The RT-qPCR of the medium samples and lysed organoid samples was consistent and no increase in viral RNA was seen. We believe that the lack of replication in the 3D organoids is likely to due to the undifferentiated state of the organoids and in this state, the organoids do not have mature enterocytes which appear to be the target cells for SARS-CoV-2 infection as seen by us and reported by Lammers et al., 2020.

Reviewer 3, comment 2

In the transcriptomics I really appreciate the honesty of the authors to report the mycoplasma contamination which most of the time unreported. However amount of the mycoplasma can be directly associated with the signaling cascade. Though authors reported DEG with or without some

samples, bit detailed analysis of the Fig S4 as more supplementary files, e.g. what are the overlap of genes, did the key 77 genes changes after changing the removal of the most significant genes?

Our response

We did not perform an extensive analysis of the DEGs after removing specific donors. However, out of 77 genes, the genes listed in table 1 related to viral infections were checked and remained significant after the analysis in Fig S4. Importantly, ATP1A1 and ATP1A1OS remained significant as well leading us to conclude that the mycoplasma contamination does not affect the downregulation of this gene.

Reviewer 3, comment 3

The top key gene ATP1A1 seems to have role in AKT pathway and it was shown earlier that SARS-CoV-2 can uses these pathways for it's replication. Please discuss about that.

Our response

ATP1A1 is an important gene that is involved in several key pathways including the AKT signaling pathway. It has been shown that ATP1A1 inhibition results in src activation and blocks src-mediated endocytic uptake. This is mentioned in the discussion on line 274-276. It is not known how exactly ATP1A1 inhibition blocks endocytic uptake but we speculate that ATP1A1 inhibition leads to src/AKT activation and downstream depletion of PIP2 from the cell surface which is necessary for endocytic entry. However, as these pathways involve highly complex interactions and as we have not performed in depth molecular analysis, we have refrained from speculating on the precise mechanisms.

Reviewer 3, comment 4

Please add limitation section in the manuscript.

Our response

Although we have included several limitations in the discussion section, we agree with the reviewer that this can be expanded upon and we have, therefore, added an additional section at the end of the discussion (lines 285-294) to highlight some methodological limitations.

Reviewer 3, comment 5

Line 180-181: It states "Some of the SARS-CoV-2 associated genes (GAS6, IFI27, SLC6A20)" Please change the terms, they are not SARS-CoV-2 associated genes, these are for many viruses e.g IFI27.

Our response

These genes are referred to as SARS-CoV-2 associated genes as our search was primarily looking at DEGs and their known association with SARS-CoV-2 infection. However, the term "SARS-CoV-2 associated" has now been removed from this line to avoid misrepresentation.

References

1. Lamers MM, Beumer J, van der Vaart J, et al. SARS-CoV-2 productively infects human gut enterocytes. *Science*. 2020;369(6499):50-54. doi:10.1126/science.abc1669
2. Zhao X, Li C, Liu X, et al. Human Intestinal Organoids Recapitulate Enteric Infections of Enterovirus and Coronavirus. *Stem Cell Reports*. 2021;16(3):493-504. doi:10.1016/j.stemcr.2021.02.009
3. Stanifer ML, Kee C, Cortese M, et al. Critical Role of Type III Interferon in Controlling SARS-CoV-2 Infection in Human Intestinal Epithelial Cells. *Cell Rep*. 2020;32(1):107863. doi:10.1016/j.celrep.2020.107863
4. van Keulen BJ, Romijn M, Bondt A, et al., Breastmilk: A Source of SARS-CoV-2 Specific IgA Antibodies (6/19/2020). Available at SSRN: <https://ssrn.com/abstract=3633123>
5. Xiaojie S, Yu L, Lei Y, Guang Y, Min Q. Neutralizing antibodies targeting SARS-CoV-2 spike protein [published online ahead of print, 2020 Dec 15]. *Stem Cell Res*. 2020;50:102125. doi:10.1016/j.scr.2020.102125

June 27, 2022

RE: Life Science Alliance Manuscript #LSA-2022-01432-TR

Dr. Adithya Sridhar
Amsterdam University Medical Centers
Meibergdreef 9
Amsterdam 1105AZ
Netherlands

Dear Dr. Sridhar,

Thank you for submitting your revised manuscript entitled "Human milk inhibits some enveloped virus infections, including SARS-CoV-2, in an intestinal model". We would be happy to publish your paper in Life Science Alliance pending final revisions necessary to meet our formatting guidelines.

- please add ORCID ID for secondary corresponding author-they should have received instructions on how to do so
- please use the [10 author names, et al.] format in your references (i.e. limit the author names to the first 10)
- please add the Twitter handle of your host institute/organization as well as your own or/and one of the authors in our system
- please consult our manuscript preparation guidelines <https://www.life-science-alliance.org/manuscript-prep> and make sure your manuscript sections are in the correct order and that your figure/table/video legends are in their own separate section
- please add a callout for your Video S1 in your main manuscript text
- in the Data Availability Statement, please include any relevant accession information for the RNA-seq data

A. FINAL FILES:

B. MANUSCRIPT ORGANIZATION AND FORMATTING:

Sincerely,

Reviewer #1 (Comments to the Authors (Required)):

The authors have addressed my concerns in a satisfactory matter

Reviewer #3 (Comments to the Authors (Required)):

The authors addressed my comments with a limitation section or modifications in the text.

July 14, 2022

RE: Life Science Alliance Manuscript #LSA-2022-01432-TRR

Dr. Adithya Sridhar
Amsterdam University Medical Centers
Meibergdreef 9
Amsterdam 1105AZ
Netherlands

Dear Dr. Sridhar,

Thank you for submitting your Research Article entitled "Human milk inhibits some enveloped virus infections, including SARS-CoV-2, in an intestinal model". It is a pleasure to let you know that your manuscript is now accepted for publication in Life Science Alliance. Congratulations on this interesting work.

DISTRIBUTION OF MATERIALS:

Again, congratulations on a very nice paper. I hope you found the review process to be constructive and are pleased with how the manuscript was handled editorially. We look forward to future exciting submissions from your lab.

Sincerely,
